# Fatty Acid Profile and *Escherichia coli* and *Salmonella* sp. Load of Wild-Caught Seaweed Fly *Fucellia maritima* (Haliday, 1838) (Diptera: Anthomyiidae)

**DOI:** 10.3390/insects15030163

**Published:** 2024-02-28

**Authors:** Felipe Lourenço, Ricardo Calado, Marisa Pinho, Maria Rosário Domingues, Isabel Medina, Olga M. C. C. Ameixa

**Affiliations:** 1CESAM—Centre for Environmental and Marine Studies, ECOMARE—Laboratory for Innovation and Sustainability of Marine Biological Resources, Department of Biology, University of Aveiro, 3810-193 Aveiro, Portugal; rjcalado@ua.pt (R.C.); marisapinho@ua.pt (M.P.); 2CESAM—Centre for Environmental and Marine Studies, Department of Chemistry, University of Aveiro, Campus de Santiago, 3810-193 Aveiro, Portugal; mrd@ua.pt; 3Mass Spectrometry Center (LAQV-REQUIMTE), Department of Chemistry, University of Aveiro, Campus de Santiago, 3810-193 Aveiro, Portugal; 4IIM-CSIC—Instituto de Investigacións Mariñas-Consejo Superior de Investigaciones Científicas, R/Eduardo Cabello Nº6, 36208 Vigo, Spain; medina@iim.csic.es

**Keywords:** insect feed, aquaculture, essential fatty acids (EFAs), lipidic profile, coastal insect species

## Abstract

**Simple Summary:**

The aquaculture industry is expected to grow in the coming years, and this means more sustainable ways are urgently needed to feed cultured animals. Insects are a promising ingredient for fish and shrimp aquafeeds, as they can convert agricultural waste into nutritious biomass. However, insect species that are currently commercially available lack some essential nutrients, such as omega-3 fatty acids, necessary for suitably growing marine organisms. We have screened the native wild seaweed fly, *Fucellia maritima*, to increase current knowledge on the nutritional diversity that insects may add to aquafeeds and for the presence of pathogenic bacteria. We found that these flies have a good amount of healthy fats, including important fatty acids that are beneficial for marine fish. Additionally, they have an acceptable amount of Enterobacteriaceae for animal feed and no presence of *Salmonella* sp. This finding suggests that *Fucellia maritima* can be a valuable ingredient for aquafeed formulation, enhancing the growth and overall health of farmed marine animals.

**Abstract:**

World aquaculture is expected to continue to grow over the next few decades, which amplifies the need for a higher production of sustainable feed ingredients for aquatic animals. Insects are considered good candidates for aquafeed ingredients because of their ability to convert food waste into highly nutritional biomass. However, commercially available terrestrial insect species lack *n-3* long-chain polyunsaturated fatty acids (LC-PUFAs), which are essential biomolecules for marine cultured species. Nevertheless, several coastal insect species feature LC-PUFAs in their natural fatty acid (FA) profile. Here, we analysed the lipidic profile of wild-caught seaweed fly *Fucellia maritima*, with a focus on their FA profile, to evaluate its potential to be used as an aquafeed ingredient, as well as to screen for the presence of pathogenic bacteria. Results showed that the flies had a total lipid content of 13.2% of their total dry weight. The main classes of phospholipids (PLs) recorded were phosphatidylethanolamines (PEs) (60.8%), followed by phosphatidylcholine (PC) (17.1%). The most abundant FA was palmitoleic acid (C16:0) with 34.9% ± 4.3 of total FAs, followed by oleic acid (C18:1) with 30.4% ± 2.3. The FA composition of the flies included essential fatty acids (EFAs) for both freshwater fish, namely linoleic acid (C18:2 *n-6*) with 3.4% ± 1.3 and alpha-linoleic acid (C18:3 *n-3*) with 3.4% ± 1.9, and marine fish, namely arachidonic acid (C20:4 *n-6*) with 1.1% ± 0.3 and eicosapentaenoic acid (C20:5 *n-3*) with 6.1% ± 1.2. The microbiological analysis found 9.1 colony-forming units per gram (CFU/g) of Enterobacteriaceae and no presence of *Salmonella* sp. was detected in a sample of 25 g of fresh weight. These findings indicate that *Fucellia maritima* biomass holds the potential to be used as an additional aquafeed ingredient due to its FA profile and the low count of pathogenic bacteria, which can contribute to the optimal growth of fish and shrimp with a low risk of pathogen transfer during the feed production chain.

## 1. Introduction

Aquaculture has grown steadily over the past several decades and is expected to grow even more by 2050 [1]. Aquaculture was responsible for the production of 88 million tonnes of aquatic animals in 2020 (49% of total production) [1]. However, most fish species produced in aquaculture are highly dependent on a diet rich in protein and long-chain polyunsaturated fatty acids (LC-PUFAs), and with the decline in the production of fish meal (FM) and fish oil (FO), new ingredients have been researched and used [1].

Among the new ingredients being explored are those that are plant-based, which are often not entirely sustainable and might contain anti-nutritional elements [2,3]. Additionally, single-cell organisms have also been tested, but these are costly, and several are yet to be readily available at a commercial scale [3,4]. Despite drawbacks, namely negative impacts on the environment and potential disease transmission risks, livestock sources have also been considered [3,5]. Another potential option to help solve this ingredient crisis are insect-based meals, although terrestrial insects are relatively deficient in *n-3* LC-PUFAs [3]. Curiously, coastal insect species are known to naturally have LC-PUFAs in their biochemical profiles [6,7], and their potential use as aquafeed ingredients remains to be fully investigated [8].

The seaweed fly *Fucellia maritima* (Haliday, 1838) (Diptera: Anthomyiidae) is an endogenous species from the European continent. It feeds on decaying organic matter, commonly termed beach wrack, that is washed upon coastal shores. Additionally, it is the only known seaweed fly species to also feed on decaying animal matter [9]. Furthermore, it is possible to rear this fly in captivity under a controlled environment [9]. This species was recently recorded for the first time on mainland Portugal, where it can be seen throughout the year overflying beach wrack [10].

The literature suggests that seaweeds can be contaminated by human pathogenic bacteria, such as *Escherichia coli* [11]. Considering that *F. maritima* feeds on decomposing seaweeds, it is possible that the flies could carry human pathogenic bacteria. Currently, insect production in Europe operates under the EU regulatory framework of good hygiene practices (GHPs). Moreover, according to the Commission Regulation (EC) 142/2011 of the European Union, insect-processed animal proteins (PAPs) are required to be tested before dispatch, where *Salmonella* sp. needs to be absent in samples of 25 g fresh weight (FW), and Enterobacteriaceae cannot exceed 300 colony-forming units (CFU) in samples of 1 g FW [12].

In this study, we performed an analysis of the lipid profile of adult *F. maritima* flies caught in their natural habitat, and in addition, we screened them for pathogenic bacteria (e.g., *Escherichia coli* and *Salmonella* sp.). Our primary objective was to evaluate the nutritional value of these insects as a source of LC-PUFAs and identify if the flies carry bacteria known to be pathogenic for humans. By investigating the lipid composition of these flies, we aimed to assess their suitability as a potential aquafeed ingredient, considering their nutritional value.

## 2. Materials and Methods

### 2.1. Insect Sampling

Adult specimens of *F. maritima* overflying beach wrack were captured using a sweeping net at Barra Beach, Aveiro, Portugal, in October 2020 (40°37′44.3″ N 8°44′42.0″ W). Collected specimens were stored in 50 mL polypropylene flasks for transportation to the insectarium facility located at ECOMARE (University of Aveiro), where they were flash-frozen at −80 °C, before being freeze-dried.

### 2.2. Sample Processing and Analysis

After being freeze-dried, the samples of adult flies were transported to the Chemistry Department (University of Aveiro), where they were grounded with a porcelain mortar and pestle. Due to the low amount of flies captured, a biomass of 10 mg was used in each replicate (n = 2) for the lipid extraction and phospholipids (PLs), whereas to analyse fatty acid (FA) content, a direct methylation of grounded adult flies was performed, with five different samples being used (n = 5).

### 2.3. Lipid Extraction, Phospholipid Identification, and Fatty Acid Analysis

The lipid extraction was performed by adapting the Folch protocol [13], using 10 mg of the sample, and adding 1 mL of ultrapure water and 3 mL of a 2:1 dichloromethane/MeOH mixture, vortexed for 30 s. Samples were then incubated on ice for 30 min, with a 30 s vortex every 5 min. Afterward, the samples were centrifuged for 5 min at 1500 rpm to separate the organic phase. The organic phase was collected using a micropipette and transferred to another tube, followed by the addition of 1 mL of a 2:1 dichloromethane/MeOH mixture to re-extract the aqueous phase; subsequently, it was vortexed and centrifuged again before transferring the organic phase to the second tube. 

After the extraction, the second tube was dried under a nitrogen stream, then resuspended with 300 μL of dichloromethane, vortexed, and transferred to a previously dried and weighted vial, repeating this process two times to transfer the total lipid extract to the vial. Afterward, the vial was dried under a nitrogen stream, which was weighted to calculate the weight of the total lipid extract, and stored at −20 °C.

Lipid classes were separated and quantified by thin-layer chromatography (TLC) analysis, according to Christie [14]. To separate PLs, 10 μg of lipid extract (n = 2) was transferred to a glass tube that was dried on a nitrogen stream. This was followed by adding 125 μL of 70% perchloric acid and heated at 180 °C for 60 min. Phosphate standards were prepared from a solution of NaH_2_PO_4_.2H_2_O with 100 μg/mL of phosphorous (P). Standards and samples were re-suspended in 125 μL of 70% perchloric acid. The samples were heated for 1 h at 180 °C in a heating block (Stuart, Staffordshire, UK). Then, 825 μL of Milli-Q water and 125 μL of 2.5% NaMoO_4_ were added. H_2_O was added to the lipid samples and vortexed again. After this procedure, 125 μL of 10% ascorbic acid was added to each sample and standard and vortexed again, after which the samples were placed in boiling water (100 °C) for 10 min. Afterward, the absorbance was measured at 797 nm in an ultraviolet–visible (UV–Vis) spectrophotometer (Multiskan GO, Thermo Scientific, Hudson, NH, USA). The amount of PLs was calculated by multiplying the quantity of determined phosphorus (μg) by 25. Two duplicates of two independent measurements were carried out for each sample. For the separation of PLs, a TLC silica plate was pre-washed in a solution of chloroform/methanol (1:1, *v*/*v*), which was then dried in a fume hood for 15 min, before being sprinkled with 2.3% boric acid, dried again in the fume hood for 15 min, and oven-dried (100 °C) for 15 min. After cooling down to room temperature, the sample containing 30 μg of PLs was applied to the silica plate and placed on a chamber saturated with chloroform/ethanol/water/triethylamine (30/35/7/35 *v*/*v*) to allow the full migration of PLs. After approximately 3 h, the plate was removed and dried under the fume hood for 20 min, then sprinkled with primuline (50 µg/mL) acetone/water (80:20), followed by drying in the fume hood, and then revealed under UV light. 

To quantify the different PL classes, the TLC spots were scratched and transferred to a glass tube. The PLs’ quantification proceeded as described before, adding a step at the end of transferring the quantification solution to an Eppendorf and centrifuging for 5 min at 1000 rpm before 200 μL of the samples was transferred to a 96-well plate to separate the PLs from the silica.

For the FA analysis, 30 μg of the samples (n = 5) was transferred to small glass tubes before adding 1 mL of methyl nonadecanoate (Sigma-Aldrich chemicals, St. Louis, MO, USA, Lot BCBQ6948V) as an internal standard, followed by vortexing after the addition of 200 μL of KOH (2M) in MeOH, before adding a saturated solution of NaCl and being centrifuged for 5 min at 2000 rpm. The samples were then dried in a nitrogen stream and resuspended with 100 μL of hexane for injection in the gas chromatography–mass spectrometry (GC-MS) procedure.

### 2.4. Microbiological Analysis

The microbiological analysis was performed in a private laboratory, following the laboratory internal methods validated by AOAC PTM.018.04 for Enterobacteriaceae enumeration of colony-forming units per gram (CFU/g), which uses the method ISO 21528-2:2017 as reference [15], ISO 16649-2:2001 for *E. coli* enumerating of CFU/g [16], and the method certificate by AFNOR BRD 07/11-12/05 for *Salmonella* sp. detection which uses the ISO 6579-1:2017 as reference [17]. A composite sample of approximately 4720 wild-caught *F. maritima* adult flies, corresponding to a biomass of 26 g FW, was analysed. Considering the small size of each specimen, the whole adult fly body was used.

## 3. Results

### 3.1. Lipid Content, Classes, and Phospholipids

The total fat content of adult flies was 13.2% of their total dry weight (DW). Total lipids are formed by triacylglycerols (TG, 60–75%), free fatty acids (FFA, 20–35%), cholesterol (1–6%), and PLs (1–4%). The TLC (Table 1 and Figure 1) allowed for the identification of the main PL classes, and the results showed that the most abundant PL class was phosphatidylethanolamine (PE), followed by phosphatidylcholine (PC).

### 3.2. Fatty Acids’ Relative Abundance

The GC-MS analysis showed that the most abundant FA was palmitoleic acid (C16:1 *n-7*), followed by oleic acid (C18:1 *n-9*). Regarding the amount of LC-PUFAs, such as eicosapentaenoic acid (C20:5 *n-3*) or arachidonic acid (C20:4 *n-6*), these were present in a considerably lower amount. The *n-6*/*n-3* ratio was 0.5 ± 0.1 (Table 2). A chromatogram with identified peaks can be visualized in Figure 2. 

### 3.3. Microbiological Analysis

The microbiological analysis revealed a total count of 9.1 CFU/g of Enterobacteriaceae and 270 CFU/g of *E. coli*, while *Salmonella* sp. was not detected in a sample of 25 g FW.

## 4. Discussion

To the best knowledge of the authors, the present study is the first-ever profiling of the lipid content of the seaweed fly *F. maritima*. The adult flies used in this study were captured in the wild overflying beach wrack, which contained mainly the brown macroalgae *Fucus* sp., but also the invasive freshwater hyacinth *Eichhornia crassipes*.

The lipid content (13.2%, DW) was considerably lower compared to insect species commercially available, such as black soldier fly (BSF) prepupae, with an average lipid content of 35.3% [18], *Tenebrio molitor* ranging from 22.3% to 30.0% [19], and slightly lower than *Musca domestica*, ranging from 16.1% to 21.2% [20]. This lower lipid content is closer to that displayed by other species of seaweed flies, such as *Coelopa frigida* (12.1% to 19.7%) and *C. pilipes* (14.2% to 16.8%) [7]. This value is also similar to other coastal fly species *Machaerium maritimae,* at 12.0% [6]. The lower levels of fats in *F. maritima* might be an indication of high protein content, as it occurs in other insects species [21]. For instance, when supplemented with a diet rich in protein, *T. molitor* shows a higher protein content with a lower fat content than when supplied with a control diet [22].

The two most abundant PLs were PEs (60.8%) followed by PC (17.1%). Usually, in terrestrial edible insects, the most abundant PL class is PC, with, for instance, values going up to 66% of total PLs in silkworm (*Bombyx mori*), and 58% in the field cricket *Gryllus assimilis* [23]. PEs are commonly most abundant in freshwater insects, like *Diamesa tonsa* and *Pseudodiamesa branickii* with 90% and 80% of total PLs, respectively [24]. Additionally, another species of Diptera, the flesh fly *Sarcophaga similis*, also displays a higher value of PEs compared to PC [25]. The same occurs in *Hermetia illucens*, black soldier fly (BSF), with up to five times more PEs than PC [26]. Species of the genus *Drosophila* also show a higher value of PEs (up to 67%) when compared to PC (17%) [27]. The previous authors suggest that the higher value of PEs is related to cold resistance adaptations, where the abundance of PEs is higher than PC in species from temperate regions in comparison to tropical regions [24,25,27]. Considering that *F. maritima*, as most members of the family Anthomyiidae, is essentially a Palearctic species, the much higher abundance of PEs compared to PC could be an adaptation to colder weather, as this is a species that is active even during the winter [10,28].

Supplementation with PEs in the diets of zebrafish, *Danio rerio*, can increase egg diameter and larval survival rate, making this phospholipid important for aquafeed formulation [29]. Moreover, the addition of PEs to the diet of the large yellow croaker (*Larimichthys crocea*) was shown to alleviate damage in intestinal cells when using a diet rich in SFAs [30]. The second most abundant PL in *F. maritima*, PC, is recognized for its role in lipid metabolism, liver function, and transport of lipids in the body. Supplementing a diet low in fish meal with PC can significantly increase lipid digestibility, subsequently eliminating excessive gut mucosal lipid accumulation in Atlantic salmon (*Salmo salar* L.) [31]. The presence of PC and PEs in high levels suggests that *F. maritima* meal may have beneficial properties related to lipid metabolism if included in aquafeeds.

The GC-MS analysis showed that the most abundant FA was palmitoleic acid (C16:1 *n-7*), which is known to have strong antibacterial activity, and its concentrated oil has been successfully used to inhibit the growth of the fish pathogen *Streptococcus agalactiae* [32]. The second most abundant FA, oleic acid (C18:1 *n-9*), is commonly found in many animal and vegetable oils, including olive oil. The role of this FA in aquaculture organisms remains mostly unexplored. However, one study found that supplementing the diet of European sea bass (*Dicentrarchus labrax*) with oleic acid can decrease feed intake and increase feed efficiency [33]. The third most abundant FA was palmitic acid (C16:0). This is the most common FA found in animals, plants, and microorganisms. When used in low concentrations, C16:0 shows great potential to reduce the mortality caused by the viral pathogen Spring Viremia of Carp Virus (SVCV), in *D. rerio* [34]. These three FAs are also the most abundant in other seaweed flies, namely *C. frigida* and *C. pilipes* [7]. 

In terrestrial insect species, such as BSF, the most abundant FAs are usually lauric acid (C12:0), a saturated fatty acid (SFA), followed by C16:0 or stearic acid (C18:0), depending on the substrate used to feed the larvae, which can influence the FA profile [35,36,37]. For instance, Ameixa et al. [38] showed that BSF fed with olive pomace displayed a higher composition of monosaturated oleic FA (18:1). In *M. domestica*, the most abundant fatty acids are C16:1 *n*-7, C16:0, or C18:1 *n*-9, also varying according to the substrate used [39], whereas in *T. molitor*, the most abundant fatty acids are oleic (C18:1 *n*-9), linoleic acid (C18:2 *n*-6), and palmitic C16:0 [19].

Among the PUFAs, alpha-linoleic acid (C18:3 *n-3*) was the most abundant, followed by C18:2 *n-6*. Both PUFAs are essential fatty acids (EFAs) for freshwater fish, as in these fishes the FA C18:3 *n-3* can be elongated and desaturated into C 20:5 *n-3* and docosahexaenoic acid (C22:6 *n-3*) [40]. Moreover, *F. maritima* possesses a C18:3 *n-3*/C18:2 *n-6* ratio close to 1, which is essential for the optimal growth of freshwater fish like the grass carp *Ctenopharyngodon idella* [41].

Concerning other biomolecules known to be EFAs for aquafeeds, the most abundant was EPA (C20:5 *n-3*), followed by arachidonic acid (C20:4 *n-6*), both LC-PUFAs. Although the supplementation of C20:4 *n*-6 in marine organisms does not necessarily affect their growth, it can increase the overall health of aquaculture animals, such as fish or sea cucumbers [42,43,44]. Moreover, supplementation with C20:4 *n*-6 can improve reproductive functions in marine fish, such as gonadal development, spawning performance, egg quality, hatching rate, and larval quality [45]. On the other hand, C20:5 *n*-3 is essential for fish growth, and our results show that *F. maritima* has a higher value of this FA than the minimum requirements for freshwater fish [46,47]. Additionally, a level of 6% of C20:5 *n*-3 is higher than the minimum requirements for the marine fish Florida pompano (*Trachinotus carolinus*) [48]. The inclusion of C20:5 *n*-3 is also important for the overall health and growth of the Atlantic salmon (*Salmo salar*) with a minimum requirement of 0.5% C20:5 *n*-3 of total FAs for normal growth [49].

The relative abundance of C20:5 *n-3* and C18:2 *n-6* is close to that recorded for *C. pilipes* and *C. frigida*; nonetheless, *F. maritima* has a lower value of C20:4 *n-6* and a higher value of C18:3 *n-3* when compared to these two species [7]. While for BSF, the values of EFAs are lower than those of *F. maritima* when using a control diet, they can increase significantly when using microalgae or expired fish feeds as feeding substrates [36,37]. Additionally, *F. maritima* has a lower n-6/n-3 ratio than *T. molitor* (0.5 and 17.8 to 64.3, respectively) [50]. The amount of n-3 LC-PUFAs in *F. maritima* is higher than any other insect species currently allowed by the EU to be produced as animal feed, namely *M*. *domestica*, *H*. *illucens*, *T*. *molitor*, *Alphitobius diaperinus*, *Acheta domesticus*, *Gryllodes sigillatus*, *G*. *assimilis*, and *B*. *mori* [51,52,53].

It is plausible that *F. maritima*, as some insect species, can biosynthesize de novo C20 LC-PUFAs, through the pathway of elongation/desaturation of C18:3 *n*-3 [54,55]. Nevertheless, it remains uncertain whether *F. maritima* and other seaweed flies use this pathway or directly assimilate LC-PUFAs from their marine dietary sources. Therefore, further studies are needed to analyse if different substrates can modulate the FA profile of this seaweed fly.

The microbiological analysis showed that wild *F. maritima* had a lower Enterobacteriaceae count than other insect species commercially available, such as mealworms (*T*. *molitor*), locusts (*Locusta migratoria*), and morio worms (*Zophobas morio*) [56]. Also, in a risk assessment from the Netherlands for locusts, lesser mealworms, mealworms, and mealworm snacks, the concentration of Enterobacteriaceae in 65% of samples exceeded the criterion for raw materials used in meat preparations (10^3^ CFU/g) [56]. However, higher counts were found in *F. maritima*, when compared to other species of flies like BSF, with 7.2 ± 0.5 CFU/g [57]. Even though only one sample was analysed, these values are in accordance with the Commission Regulation (EC) 142/2011 of the European Union for insect PAPs, making this species a possible candidate for aquafeed ingredients [12].

In conclusion, the FA profile of this seaweed fly in the wild presents many biomolecules of interest for the formulation of aquafeeds, whether for the maintenance of optimal growth of aquatic animals or the overall health of farmed fish, considering this species’ high level of C20:5 *n-3*, an essential FA for optimal fish and shrimp growth. Moreover, *F. maritima* contains highly valuable PLs, which are important supplements for aquafeed, especially in early life stages [58,59]. Additionally, these flies display lower counts of human pathogenic bacteria when compared to other insect species already used in commercial applications, and their use may contribute towards the diversification of insect species production for feed, thus enhancing the resilience of this young industry [8]. 

However, to fully assess the suitability of *F. maritima* as a viable aquafeed ingredient, it is imperative to establish standardized rearing protocols that can be manipulated and replicated for industrial purposes. In fact, we have already started rearing experiments as detailed in Lourenço et al. [60]. Furthermore, we recommend additional research exploring various substrates, such as fish by-products and diverse seaweed species prevalent in beach wrack. The latter is often considered a marine waste, leading to its disposal in landfills, incurring on public expenses, and contributing to environmental degradation [61]. By using different substrates to culture seaweed flies under controlled conditions, it is necessary to analyse if their protein content and lipid profile are affected. Moreover, as insects’ larvae are more commonly used for industrial applications, it is also necessary to analyse in the future the nutritional profile of *F. maritima* larvae. Considering the pressing need for sustainable marine aquafeed alternatives, understanding the viability and availability of this potential ingredient is paramount for advancing the expansion of sustainable aquaculture.

## Figures and Tables

**Figure 1 insects-15-00163-f001:**
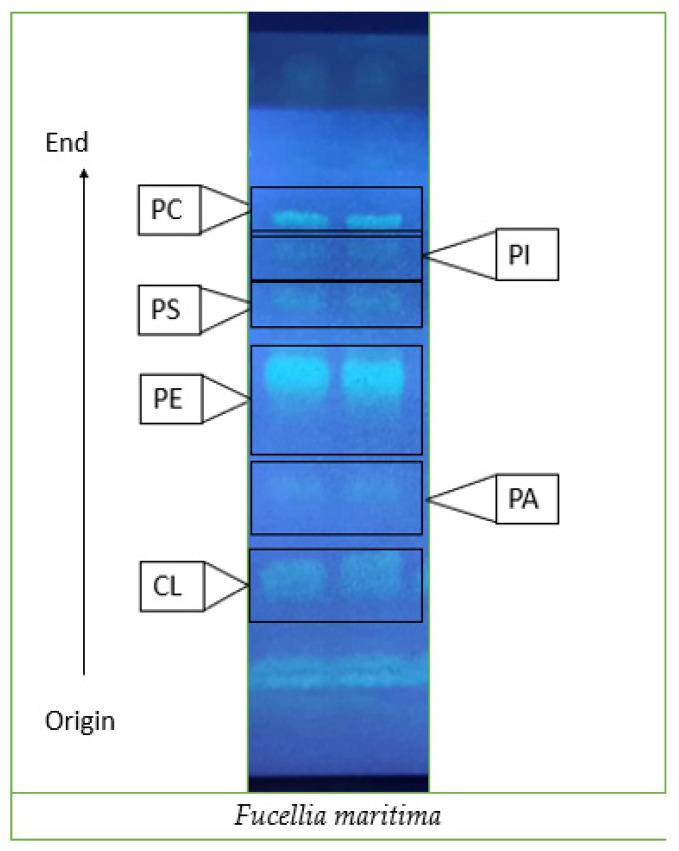
Separation of phospholipids of adult *Fucellia maritima* collected from the wild by thin-layer chromatography using a silica plate.

**Figure 2 insects-15-00163-f002:**
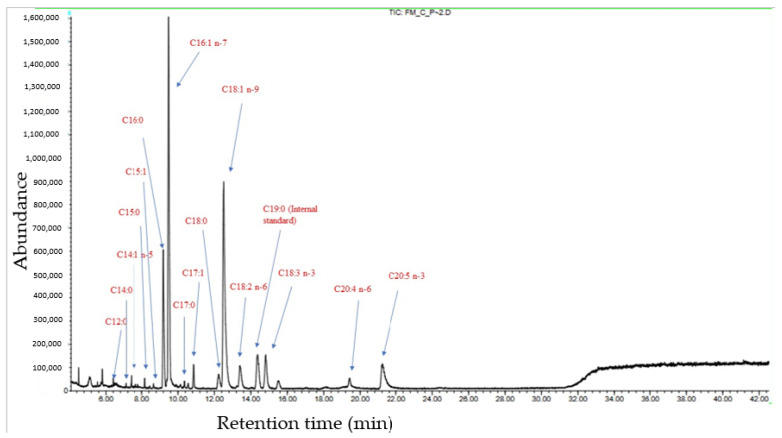
GC-MS chromatogram with the major fatty acids detected in adult *Fucellia maritima* collected from the wild.

**Table 1 insects-15-00163-t001:** Quantification of phospholipids. Data are expressed as mean values of two replicates.

	μg/mg Lipid	% of Total Phospholipids
Cardiolipin (CL)	4.9	8.3
Phosphatidic acid (PA)	1.1	2.0
Phosphatidylcholine (PC)	10.0	17.1
Phosphatidylethanolamine (PE)	35.7	60.8
Phosphatidylinositol (PI)	3.4	5.8
Phosphatidylserine (PS)	3.4	5.8

**Table 2 insects-15-00163-t002:** FA profile of *Fucellia maritima*, quantified by GC-MS, expressed as relative abundance (%). Values are the means of five replicates ± standard deviation (SD).

Fatty Acids	Relative Abundance (%) ± SD
C12:0 (Lauric acid)	0.0 ± 0.0
C14:0 (Myristic acid)	0.1 ± 0.4
C15:0 (Pentadecanoic acid)	0.4 ± 0.1
C16:0 (Palmitic acid, PA)	14.9 ± 1.9
C17:0 (Heptadecanoic acid)	0.2 ± 0.1
C18:0 (Stearic acid)	2.4 ± 0.5
SFA	18.8 ± 0.7
C14:1 *n-5* (Myristoleic acid)	0.9 ± 0.1
C15:1 (Pentadecanoic acid *cis*-10)	0.1 ± 0.1
C16:1 *n-7* (Palmitoleic acid)	34.9 ± 4.3
C17:1 (Heptadecanoic acid *cis*-10)	1.2 ± 0.4
C18:1 *n-9* (Oleic acid)	30.4 ± 2.3
MUFA	66.9 ± 1.2
C18:2 *n-6* (Linoleic acid)	3.4 ± 1.3
C18:3 *n-3* (α-Linolenic acid)	3.4 ± 1.9
C20:4 *n-6* (Arachidonic acid)	1.1 ± 0.3
C20:5 *n-3* (Eicosapentaenoic acid)	6.1 ± 1.2
PUFA	14.0 ± 1.1
*n-6* PUFAs	4.5 ± 0.9
*n-3* PUFAs	9.5 ± 1.2
*n-6*/*n-3* ratio	0.5 ± 0.1

## Data Availability

All raw data of relative abundance (%) of fatty acids (FAs) analysis, and phospholipids quantification available as Appendix A.

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
