# Peer review of "Fatty Acid Profile and Escherichia coli and Salmonella sp. Load of Wild-Caught Seaweed Fly Fucellia maritima (Haliday, 1838) (Diptera: Anthomyiidae)"

_insects, 2024, doi:10.3390/insects15030163_

Round 1

Reviewer 1 Report

Comments and Suggestions for Authors

The manuscript investigates the potential of Seaweed Fly in aquaculture feed, analyzing its fatty acid composition and screening for pathogens. Overall, the manuscript demonstrates good scientific content and data accuracy but could benefit from improvements in linguistic expression.

The manuscript presents a compelling investigation into the potential of Seaweed Fly in aquaculture feed, with a focus on its fatty acid composition and pathogen screenings. While the scientific content and data accuracy are commendable, the manuscript would benefit from expanded content and improved clarity in certain areas:  

1. I recommend transforming the content into a full article if possible. Specifically, the "2.2 Sample processing and analysis" section on page 3 (line 100) could be further divided into three sub-sections: sample processing, lipid TLC analysis, and microbiological analysis. The latter should provide a detailed description of the methodology, specifically, which tissues were used for cultivation and how the samples were treated (page 4, line 152).  

2. The sample processing portion left me curious. The choice to use five adult individuals as a single sample (page 3, line 103) and to only prepare two samples is puzzling. Why not use weight as a universal standard between samples? Two samples are insufficient to calculate a standard deviation. Further, it is unclear whether five repeats (n=5) for fatty acid (FA) content analysis were randomly drawn from each extract or taken from the two samples in total. This section should be rewritten for clarity.  

3. The results section (page 4, line 157) should be divided and described with subheadings, such as lipid content and types, GC-MS, and microbiological culture results. The GC-MS section could be enriched with additional figures. The microbiological culture section, with only one sentence of results (page 5, line 170), does not provide comprehensive information. I suggest adding photographs or other solid evidence. Might it also be beneficial to include an analysis of bacterial diversity?  

4. If Seaweed Fly is to be used as an additive in fish feed, what would be its stable source? Would it be artificially bred? If so, would the cost be high? It would be beneficial to address these considerations in the discussion section.    

Comments on the Quality of English Language

While generally fluent, the language used in the manuscript can be complex in some sections, potentially hindering comprehension. Simplifying and splitting longer sentences could enhance readability. There are minor grammatical errors, such as inconsistencies in number agreement and verb tense. A thorough proofreading and revision are recommended. Consistency in the usage of specific technical terms, especially when referring to types of fatty acids, is required to maintain professional standards throughout the manuscript.

Author Response

Dear reviewer,

We thank you for all the comments, suggestions and corrections as they greatly improved the overall quality of the work. Please find attached the document with the replies.

Reviewer 2 Report

Comments and Suggestions for Authors

The manuscript entitled Fatty acid profile and Escherichia coli and Salmonella sp. load of wild-caught seaweed fly Fucellia maritima (Haliday, 1838) (Diptera: Anthomyiidae) by Lourenço et al., is a simple and descriptive paper. The introduction could be improved, the methodology is short but well written, the results are well structured, and the discussion addresses several aspects of their nutritional value from an aquaculture point of view. There are some observations and questions to be answered to improve the quality of the writing.

As the authors mention about the null presence of LC-PUFA (EPA, DHA and ARA) in insects associated with offshore areas and the requirement of these fatty acids is high and a topic of major importance in aquafeeds, however, there is the question of why focus the research only on the use of this insect for aquaculture feeds? Also, the journal is for insects, could the focus be broader?

There is no justification for the introduction of a harvest potential that supports this species for the generation of a meal industry. This point is associated with the above, the search for new food sources also depends on the viability and availability of this.

Line 87 and 88 are materials and methods. Remove!

Why not report the other proximate parameters (protein, ash, ELN)?  Insects generally have a very complete amino acid profile and their potential for substitution or inclusion in diets for marine organisms is linked to the lack of LC-PUFA in their profile. If this species is low in lipids compared to other species, then its protein is high? The chitin issue is associated with this protein issue and its overestimation. Couldn't these issues be brought into the discussion?

Line 225. According to the most current references on the subjectC18:3 n3 and C182 n6 are not LC-PUFA, they are PUFA. While EPA, DHA and ARA are LC-PUFA.

Line 244-252- This paragraph is confused by the use of LC-PUFA (I suppose you refer to C18 FA). In addition, the presence of EPA and ARA in F. maritima as well as in other sea flies is not clear if their presence is due to accumulation of their food or if they could have the capacity of biosynthesis from PUFA? Do you know anything about this issue?

Author Response

Dear reviewer,

Thank you for all the comments, suggestions and corrections made, as they greatly improved the overall quality of the work. Please find attached the replies to the comments.

Reviewer 3 Report

Comments and Suggestions for Authors

The manuscript “Fatty acid profile and Escherichia coli and Salmonella sp. load of wilt-caught seaweed fly Fucellia maritima (Haliday, 1838)(Diptera: Anthomyiidae)" is interesting as it investigates the fatty acid composition of a new insect for potential use in feed. However, it would have been more interesting to also consider the larvae, as they are the most interesting stage for the feed and probably for the fat content (it is a suggestion to continue with the investigations). Furthermore, considering the final objective, a greater comparison of the results is also appropriate with other insects currently bred for the feed (not only Hermetia, but also Tenebrio and Musca). It is suggested to limit self-citations to the essential ones.

The manuscript may be accepted with minor revisions.

Below are the specific comments:

INTRODUCTION

Line 75: you rewrite “maritima” in italics

Line74-75: please delete the sentence (the sentence is uninformative and superfluous).

Line 87-88: please move this part to M&M (for example to (line 119)

MATERIALS AND METHODS

Line 104: add “(PLs)” after  “phospholipid”.

Line 148: please, it is necessary define “internal standard”

RESULTS

Line 161: please, delete “gathered”

DISCUSSION

Line 181-182: the authors limit the comparison of lipid content to Hermetia. Considering the purpose, it is necessary to add citations for other species (Tenebrio and Musca).

Line 185: also for PLs, compare insects bred for feed (for examples: Hermetia https://doi.org/10.1016/j.seppur.2020.118040; Musca https://doi.org/10.1016/S1096-4959(02)00032-5)

 Line 219-224: it is appropriate to add the comparison with the fatty acid composition of Tenebrio and Musca

Line  248: please, also add the example in Tenebrio (see  doi.org/10.3390/insects14110854)

Comments on the Quality of English Language

The English language requires minor editing

Author Response

Dear reviewer,

Thank you for all the comments, corrections and suggestions made, as they greatly improved the overall quality of the work. Please find attached the replies to all the comments.

Round 2

Reviewer 2 Report

Comments and Suggestions for Authors

The answers generated are sufficient to continue with the manuscript process.